# Outcomes of the KC life 360 intervention: Improving employment and housing for persons living with HIV

**Joseph S. Lightner**[1,2] *, **Travis Barnhart**[2], **Jamie Shank**[2,3], **Debbie Adams**[2], **Ella Valleroy**[1‡], **Steven Chesnut**[1‡], **Serena Rajabiun**[4‡]

**1** School of Nursing and Health Studies, University of Missouri-Kansas City, Kansas City, Missouri, United States of America, **2** HIV Services, Kansas City Health Department, Kansas City, Missouri, United States of America, **3** Organizational Empowerment, LLC, Atlanta, Georgia, United States of America, **4** Zuckerberg College of Health Science, University of Massachusetts Lowell, Lowell, Massachusetts, United States of America

☯ These authors contributed equally to this work.
‡ EV, SC and SR also contributed equally to this work.
* lightnerj@umkc.edu

**Data Availability Statement:** The data were submitted within the Supporting information files.

**Funding:** KCHD:89HA00028 Special Projects of National Significance (SPNS) Initiative, Improving

## Abstract

Housing and employment are key factors in the health and wellbeing of persons living with HIV (PLWH) in the United States. Approximately 14% of low-income PLWH report housing instability or temporary housing, and up to 70% report being unemployed. The purpose of this study was to examine the outcomes of an intervention to improve housing and employment for PLWH in the Midwest. Participants (N = 87) were recruited from the Kansas City metropolitan area to participate in a one-year intervention to improve housing and employment. All individuals were living with HIV and were not stably housed, fully employed, nor fully engaged in HIV medical care. A series of generalized estimating equations were conducted using client-level longitudinal data to examine how housing, employment, viral load, and retention in care changed over time. Housing improved from baseline to follow-up, with more individuals reporting having stable housing (OR = 23.5; $p < 0.001$). Employment also improved from baseline to follow-up, with more individuals reporting full-time employment (OR = 1.9; $p < 0.001$). Viral suppression improved from baseline to follow-up, with more individuals being virally suppressed (OR = 1.6; $p < 0.05$). Retention in care did not change significantly from baseline to follow-up (OR = 0.820; $p = 0.370$). Client navigation seems to be a promising intervention to improve housing and employment for PLWH in the Midwest. Additional research is needed on the impact of service coordination on client-level outcomes. Future studies should be conducted on the scalability of client navigation interventions to improve the lives of low-income, underserved PLWH.

## Introduction

In the United States, there were an estimated 1.2 million persons living with HIV (PLWH) in 2018, with about 13% unaware of their HIV status [1]. Of those in the Ryan White HIV/AIDS

Health Outcomes Through the Coordination of
Supportive Employment & Housing Services,
Health Resources and Services Administration, U.
S. Department of Health and Human Services The
funder did not play any role in the study design,
data collection or analysis, decision to publish, or
preparation of the manuscript.

**Competing interests:** The authors have declared
that no competing interests exist.

Program, 14% were unstably or temporarily housed [2], up to 70% were unemployed, (3) and 81.4% were at or below the federal poverty level [3]. The highest rates of HIV diagnoses, across sex and age groups, are from census tracts where 18% or more of the residents were living below federal poverty levels and in census tracts where median household income was below $42,000 a year [4].

An estimated 29,100 individuals were diagnosed with HIV in 2018 with only 67.9% reaching viral suppression within 6 months [4]. The United States has not reached the "90-90-90" goal put forth by UNAIDS, which aims to have 90% of PLWH aware of their status, 90% of those aware of their status receiving antiretroviral therapy, and 90% of those receiving antiretroviral therapy virally suppressed [5]. It is clear that with only 74% of PLWH retained in HIV medical care [6] the goals of "90-90-90" are not being met. New approaches that focus on other factors and social determinants of health are needed in the efforts to treat HIV.

Housing has been identified as a determinant for HIV health outcomes [3]. Inadequate and unstable housing have been linked to worse HIV health outcomes, while controlling for healthcare factors and individual patient differences [7]. Those who are homeless or unstably housed are less likely to receive care throughout diagnosis and treatment [7]. Inversely, PLWH who are receiving housing assistance or services are more likely to receive primary care health services, including HIV follow-up visits [7].

Homelessness and unstable housing have also been linked with overall poorer health outcomes for PLWH [7]. For example, homeless or unstably housed PLWH are more likely to have poorer mental or physical health functioning, more mental health diagnoses, and overall poorer quality of life. In terms of comorbidities, PLWH who are homeless or unstably housed often experience higher rates of other infections, such as hepatitis C and tuberculosis [7].

Employment status is an important predictor of health along the entire HIV care continuum [8]. Lack of employment is associated with lack of testing for HIV, and may delay diagnosis [8]. For those with an HIV diagnosis, unemployed individuals are twice as likely to miss their HIV-focused appointments than individuals who are employed [8]. Among PLWH who are unemployed, 40% report a desire to work [9].

There is emerging evidence that suggests client navigation and care coordination interventions can improve housing stability and HIV health outcomes for PLWH who are experiencing homelessness [10, 11]. In a national study of PLWH experiencing homelessness and co-occurring mental health and substance use disorders, participants who received navigation services to obtain stable housing increased their viral suppression rates from 49% to 77% [10]. Furthermore, there seemed to be a dose-response relationship between client navigation services and retention in care, such that more navigation services increased the likelihood of being retained in care [10]. Critical tasks performed by navigators included: addressing stigma, linking to housing support and search services, connecting to behavioral health care and communicating with providers and landlords to support retention in housing and medical care [12]. In addition, one study found that integrating employment services into housing programs can also support PLWH gain employment, housing stability, and improve health outcomes [13].

Several authors have highlighted the need for interventions to address homelessness and unemployment for PLWH [14, 15]. Therefore, the purpose of this study is to examine the outcomes of an intervention that aimed to increase housing and improve employment for at-risk, low-income PLWH experiencing housing instability and unemployment. We hypothesize that client navigation will be help promote improvements in employment, housing, HIV viral suppression, and retention in HIV medical care.

## Methods

### Description of the intervention

KC Life 360 is an initiative to increase employment and housing services for PLWH by providing direct client navigation and improving system-level coordination. The KC Life 360 intervention was implemented by Kansas City Health Department (KCHD). KCHD receives funding from the Ryan White HIV/AIDS Program (RWHAP) Part A (prime recipient), Part B (sub-recipient), and Housing Opportunities for Persons with AIDS (HOPWA) in partnership with community agencies: Catholic Charities of Kansas City and St. Joseph, Missouri, focused on employment services, and reStart, a housing provider focused on temporary and permanent housing. The intervention consisted of three components: employment navigation, housing navigation, and system coordination.

### Employment navigation

After baseline data collection, clients were introduced to an employment navigator who served as the primary contact and led participants through the intervention. Once enrolled, participants completed a readiness for employment assessment and participants and employment navigators developed an individualized employment plan together. The employment navigator, in collaboration with staff from Catholic Charities, provided support for clients who needed identification documents, clothing, emergency housing assistance, hygiene kits, transportation, certifications (e.g., food handler's certification from the local health department, commercial driver's license), cell phone payment, bicycles, holiday meals, legal name change for transgender clients, etc. Employment navigators met with clients weekly until they obtained employment. After employment was obtained, employment navigators met with clients as needed to assist with happiness of employment and potentially different or additional employment.

### Housing navigation

Clients also received access to a housing navigator who provided transportation, emergency assistance (e.g., food, clothing, cellphone), short-term housing assistance (e.g., unpaid rent, eviction costs, other barriers to securing permanent housing), furniture, and other items necessary for a place to live. Housing navigators met with clients weekly until they were stably housed and continued to meet with them as needed after achieving housing stability.

### System coordination

System-level changes were conducted as part of this project. First, employment and housing navigators were co-located at both service sites so that clients could access services at one time. Second, employment and housing data were added to medical case management data systems to improve coordination between providers. Employment and housing navigators were provided access to this shared system and were trained on how to upload documents (e.g., apartment lease, employment records, etc). Third, employment and housing navigators were included in monthly service coordination meetings. Fourth, medical case managers were trained on the beneficial impacts of housing and employment on their clients' health. Fifth, all KC Life 360-funded staff positions across all agencies documented client encounters (e.g., office visit, benefits coaching, job placement, employment preparation, housing case management, and more) and service referrals on the shared system. Case notes captured client housing status, clinical care (medical visits, lab draws, date of diagnosis) and employment. Case notes were used for screening, eligibility, and tracking. Lastly, electronic records allowed for

efficient reporting on client demographics, utilization, and outcomes to local HIV service organizations, as well as state and federal agencies.

A detailed program description of the intervention has been published previously [16].

### Recruitment

PLWH in the metropolitan Kansas City Ryan White program were recruited via medical case managers and housing providers to participate in a one-year intervention to improve housing and employment opportunities. HIV case managers and housing providers discussed the study with their clients and referred them to the study staff.

### Participants

To be eligible for this intervention, clients must have been: 18 years of age or older and living with HIV. unemployed or under-employed (e.g., not having enough money to meet daily needs or not having a job that used all of their skills); literally homeless at imminent risk of homelessness, unstably housed, or feeling domestic violence; and have at least one HIV health risk factor from the following: have a viral load above 200 copies/ml; diagnosed within the last 12 months; or out of care for at least 6 months.

### Data collection

Data were collected from the participants and clinical records between May 2018 and August 2020. Self-reported data were collected via in-person or telephone interviews with trained study staff. Clinical data were collected via medical records. Consent and baseline data collection occurred after clients were referred to the program by case managers and housing providers. Once enrolled in the study, participants were asked a series of questions about employment, housing, medical history, medical care, substance use, trauma, stigma, and more. Clients were interviewed at baseline, 6 months, and 12 months. Medical records were assessed retrospectively for HIV viral load and retention in care. Participants received a $25 gift card at the completion of the interview.

### Measures

**Employment.** Clients were asked, "Are you currently employed, either part-time or full-time?" Clients could answer yes or no. If clients answered yes, they were asked to describe their employment. Responses included: full-time (35 hours/week or more), part-time (less than 35 hours/week), temporary jobs (daily, weekly, or monthly throughout the year), working on a per diem cash basis, under the table, or other.

**Housing.** Clients were asked, "How would you describe your current housing situation?" Responses included: homeless, imminent risk of losing housing, unstably housed or at risk of losing housing, or stably housed. Homeless was defined as lacking a fixed, regular, and adequate nighttime residence. Homeless could include living on the streets, in a car, bus, park, abandoned building, campground, or in a temporary shelter for the homeless. Imminent risk of losing housing was defined as imminently losing their primary nighttime residence in the next 14 days with no subsequent residence identified and lacking individual or family supports to obtain permanent housing. Unstably housed or at risk of losing housing was defined as not having a lease, ownership interest, or occupancy agreement in a permanent and stable housing situation in the last 60 days; or in permanent housing but receiving a shut off notice in the last 60 days; or moved twice in the last 60 days and expecting to move again in the foreseeable future; or received an eviction notice; or fleeing domestic violence.

**Viral suppression.** Viral suppression was assessed through lab data obtained from the individual's medical record. Baseline viral suppression data were collected as the last known viral load results prior to enrollment. If the test reported HIV viral load below 200 copies/ml or undetectable, the individual was considered virally suppressed. If the lab value was above 200 copies/ml, the individual were not considered virally suppressed. Viral suppression for 6- and 12-month follow-up were reported from the last known viral load between the study timepoints.

**Retention in care.** Retention in care was assessed at baseline to capture the last primary care visit for HIV prior to enrollment in the intervention recorded in the individual's medical records. If the individual had a HIV primary care visit within six months of enrollment, they were identified as being retained in care. However, if they did not have a HIV primary care visit within 6 months of enrollment, they were identified as being out of care. Six-month and 12-month follow-up data assessed retention in care as having a HIV primary care visit within 3 months prior to 6- and/or 12-month interview dates, at least 90 days apart. If the individual did have a HIV primary care visit, they were considered retained in care. If not, they were considered not retained in care.

**Other measures.** Several aspects were measured and presented to describe the sample. Gender was assessed using a 1-item measure asking, "What is your current gender identity?" Possible responses included male, female, transman, transwoman, gender queer, gender non-conforming, or participants were able to describe their gender as they chose. Sexual orientation was assessed using a 1-item measure asking, "What is your sexual orientation?" Possible responses included heterosexual/straight, lesbian/gay/homosexual, bisexual, or other. Race was collected by asking if participants identified as White, Black/African American, American Indian/Native American, Alaska Native, Pacific Islander, Asian, or other. Ethnicity was assessed by asking if participants identified as Hispanic, Latino/a, or Spanish origin. Education was assessed by asking, "What is the highest level of education that you have completed?" Possible responses included no formal education, middle school, less than high school, high school (or GED), some junior college, junior (2-year) college, technical school, some college, college (4-year) graduate, more than 4-year college. To assess incarceration history, participants were asked if they had ever been to jail or prison. Addiction severity was assessed using the World Health Organization's ASSIST tool [17]. Depression risk was assessed using a 10-item Center for Epidemiologic Studies Depression (CES-D) Scale with higher scores indicating more risk of depression [18]. Food security in the last 6 months was assessed using a 6-item Household Food Security Scale with higher scores indicating more food insecurity [19]. Total unmet needs were assessed for 12 unmet needs that included food, clothing, housing, transportation, financial, interpreter assistance, substance use treatment, mental health treatment, legal assistance, medication assistance, job training, and dental care. The need for substance use treatment was assessed by asking participants if they needed treatment for substance use and were not able to receive it. Lifetime exposure to trauma was assessed using the Brief Trauma Questionnaire [20].

## Statistical analysis

Univariate statistics were calculated for all study variables.

We operationalized the longitudinal variables in the following ways: For housing status, the responses were treated as ordinal in which (1) = literally homeless, (2) = imminent risk of losing housing, (3) = unstably housed or at risk of losing housing, (4) = stably housed. We operationalized employment status as both a dichotomy (unemployed / employed) and as an ordinal variable. For ranked employment status, the options were treated as ordinal in

which (1) unemployed, (2) per diem cash / under table cash, (3) temporary, (4) part-time, (5) full-time. For viral suppression, we operationalized the data as an ordered dichotomy in which (0) not virally suppressed, (1) virally suppressed. For retention in care, we operationalized the data as an ordered dichotomy in which (0) not retained in care, (1) retained in care.

The *geepack* library [21] in R [22] was used to conduct binary and ordinal logistic generalized estimating equations (GEE) to understand how outcome variables changed during the one-year intervention. An intercept only model was first specified to determine the overall probability distribution (i.e., likelihood of event occurrence for binary outcomes, proportional distributions for ordinal outcomes). We then conditioned the model, specifying time as the only predictor of change. In the conditional GEE models, time was operationalized as measurement waves (time 1, 2, 3), and by observed month of measurement (month 0, 6, 12) to provide two different interpretations of time. With time operationalized by measurement wave, interpretations are better aligned with the data collection procedure and the odds of change (i.e., odds ratios) are in 6-month units. With time operationalized as measurement month, interpretations are extrapolated to the time between measures where odds of change are in 1-month units. We report on the findings using the measurement wave frame of reference; however, we report the findings using the measurement month frame of reference in our supplemental tables. All study procedures were approved by the Institutional Review Board at the University of Missouri-Kansas City (Protocol #2016108). Verbal consent was obtained from study participants.

## Results

Table 1 summarizes the demographic details of the participants in our study collected at the beginning of the intervention. On average, the sample was 35.6 years (SD: 11.7), had been living with HIV for 8.2 years (SD: 7.7), had a yearly household income of $7,589.00 (SD: $10,589.6), was mostly male (75.9%), non-Hispanic Black (56.3%) or non-Hispanic White (20.7%), and had a high school degree or less (58.6%). While the largest category of individuals reported lesbian, gay, or homosexual (44.8%) as their sexual orientation, 31.0% reported straight, 18.4% reported bisexual, and 4.6% reported other. Twenty-three individuals (26.5%) reported having a history of incarceration. The average participant reported high levels of addiction severity (18.4, SD:16.9) and depression (15.6, SD: 8.0) and very low food security (62.1%). Out of 12 unmet needs, the average individual reported that they had three needs that were unmet (SD: 2.2). However, 80.5% and 89.7% reported that they either did not need or had already received mental health services and substance use services, respectively. Trauma was high in this sample, with the average individual reported a trauma score of 4.1 (SD: 2.3).

Table 2 presents the results of housing, employment, viral suppression, and retention in care at baseline, 6-, and 12-month. At baseline, the majority of individuals were literally homeless (50.6%) or unstably housed or at risk of losing housing (42.5%) and most were unemployed (75.6%). A large portion of the sample was virally suppressed at baseline (67.5%). Approximately 1/3 of the sample (35.3%) was retained in care at baseline, with a similar proportion (31.5%) being retained at 12 months.

Housing at 6 and 12 months improved, with fewer individuals reporting literally homeless (22.2% and 5.3% at 6 and 12 months, respectively) and more reporting having stable housing (22.2% and 73.7% at 6 and 12 months, respectively). Employment also improved from baseline to follow-up (41.3% and 44.4% at 6 and 12 months, respectively), with more individuals

**Table 1. Demographic statistics (N = 87).**

|  | N or Mean | % or SD |
|---|---|---|
| Age (years) | 35.6 | 11.7 |
| Years Living with HIV | 8.2 | 7.7 |
| Yearly Household Income (USD) | $ 7,589.00 | $ 10,589.60 |
| Social Security Insurance/Disability Insurance |  |  |
| Receiving | 4 | 4.6% |
| Not receiving | 83 | 95.4% |
| Gender |  |  |
| Transgender or Other | 3 | 3.4% |
| Female | 18 | 20.7% |
| Male | 66 | 75.9% |
| Sexual Orientation |  |  |
| Bisexual | 16 | 18.4% |
| Lesbian/Gay/Homosexual | 39 | 44.8% |
| Heterosexual | 27 | 31.0% |
| Other | 4 | 4.6% |
| Race |  |  |
| Hispanic | 9 | 10.3% |
| Non-Hispanic Black | 49 | 56.3% |
| Non-Hispanic White | 18 | 20.7% |
| Other | 11 | 12.6% |
| Education |  |  |
| 4-year Degree or Beyond | 5 | 5.7% |
| Some College/2-year Degree/Technical School | 31 | 35.6% |
| High School | 31 | 35.6% |
| Less than High School | 20 | 23.0% |
| Incarceration History |  |  |
| Yes | 23 | 26.5% |
| No | 64 | 73.6% |
| Addiction Severity Score | 18.4 | 16.9 |
| Depression Score | 15.6 | 8.0 |
| Food Security |  |  |
| High or Marginal | 19 | 21.8% |
| Low | 14 | 16.1% |
| Very Low | 54 | 62.1% |
| Total Unmet Needs | 3 | 2.2 |
| Mental Health Unmet Needs |  |  |
| Met or not needed | 70 | 80.5% |
| Not met | 16 | 18.4% |
| Substance Use Unmet Needs |  |  |
| Met or not needed | 78 | 89.7% |
| Not met | 8 | 9.2% |
| Lifetime Exposure to Trauma | 4.1 | 2.3 |

Note: SD = Standard deviation, USD = United States Dollar.

reporting full-time employment (27.0% and 30.2% at 6 and 12 months, respectively). Viral suppression increased at follow-up, with 81.5% of the sample being virally suppressed at 12-month follow-up. Retention in care decreased from baseline to 6-months (35.3% to 12.6%) but increased from 6-months to 12-months (12.6% to 31.5%).

**Table 2. Baseline, 6-month, and 12-month outcomes.**

| | Baseline | | 6-month | | 12-month | |
|---|---|---|---|---|---|---|
| | **N** | **%** | **N** | **%** | **N** | **%** |
| Housing | | | | | | |
| Literally Homeless | 44 | 50.6% | 10 | 22.2% | 2 | 5.3% |
| Imminent Risk of Losing Housing | 6 | 6.9% | 2 | 4.4% | 1 | 2.6% |
| Unstably Housed or at Risk of Losing Housing | 37 | 42.5% | 23 | 51.1% | 7 | 18.4% |
| Stably Housed | 0 | 0.0% | 10 | 22.2% | 28 | 73.7% |
| Employment | | | | | | |
| Yes | 21 | 24.4% | 26 | 41.3% | 24 | 44.4% |
| No | 65 | 75.6% | 37 | 58.7% | 30 | 55.6% |
| Employment | | | | | | |
| Full-Time (35 hours/week or more) | 7 | 8.1% | 17 | 27.0% | 16 | 30.2% |
| Part-Time (Less than 35 hours/week) | 7 | 8.1% | 6 | 9.5% | 4 | 7.5% |
| Temporary Job | 4 | 4.7% | 3 | 4.8% | 1 | 1.9% |
| Per Diem Cash | 1 | 1.2% | 0 | 0.0% | 0 | 0.0% |
| Under the Table | 2 | 2.3% | 0 | 0.0% | 2 | 3.8% |
| Unemployed | 65 | 75.6% | 37 | 58.7% | 30 | 56.6% |
| Viral Suppression | | | | | | |
| Yes | 58 | 67.4% | 55 | 86.0% | 44 | 81.5% |
| No | 28 | 32.6% | 9 | 14.0% | 10 | 18.5% |
| Retention in Care | | | | | | |
| Yes | 30 | 35.3% | 11 | 12.6% | 17 | 31.5% |
| No | 55 | 64.7% | 76 | 87.4% | 37 | 68.5% |

Results (presented in full in S1–S5 Tables) show significant improvements in housing, employment, and viral suppression throughout the intervention. Retention in care did not change significantly during the one year of this intervention.

## Housing

Housing improved significantly for participants in this intervention. Fig 1 presents the proportion of individuals in each housing category at baseline, 6- and 12-months. The model for housing status by measurement wave (S1 Table) indicated that Time was a statistically significant positive predictor of housing status. At each subsequent measurement wave (i.e., 6-month, 12-month), study participants were approximately 23.5 times ($p < 0.001$) more likely to to have improved housing status, identified as being at least one rank above the previous measurement period.

## Employment status

Employment improved significantly for participants in this intervention. Fig 2 presents the proportion of individuals in each employment category at baseline, 6-, and 12-months. The model analyzing ordinal employment status by measurement wave (S2 Table) indicated that Time was a statistically significant positive predictor of employment. At each subsequent measurement wave (i.e., 6-month, 12-month), study participants were approximately 1.9 times ($p < 0.001$) more likely to increase employment status, identified as being at least one rank above the previous measurement period.

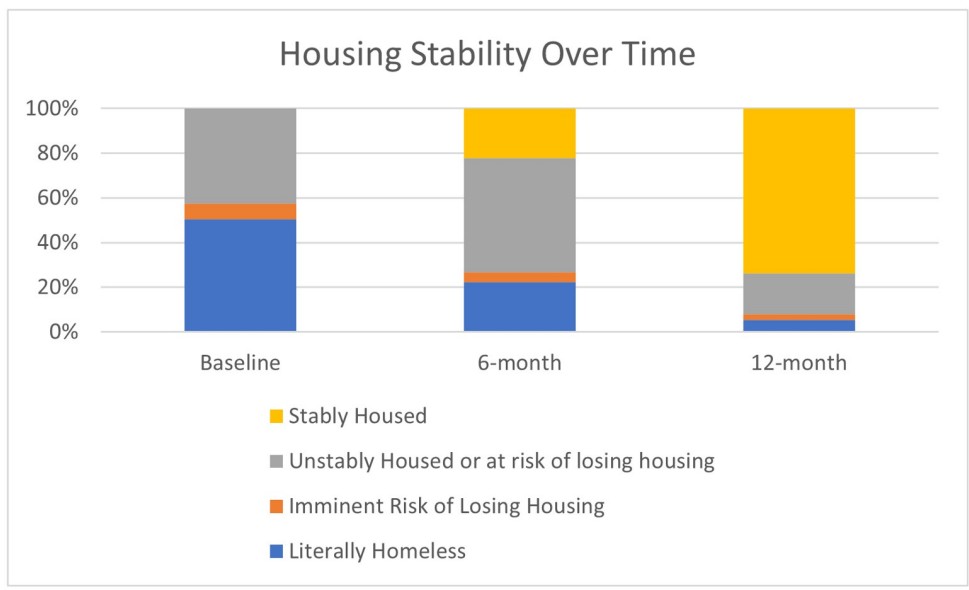

**Fig 1. Housing stability.** At each subsequent measurement wave, participants were approximately 23.5 times ($p<0.001$) more likely to be in a better housing status.

Time was also a statistically significant, positive predictor of the binary measure of employment (S3 Table). At each subsequent measurement wave, study participants were approximately 1.59 times ($p = 0.01$) more likely to be employed.

## Viral suppression

Viral suppression improved significantly for participants in the intervention. The model analyzing viral suppression by measurement wave (S4 Table) indicated that time was a statistically

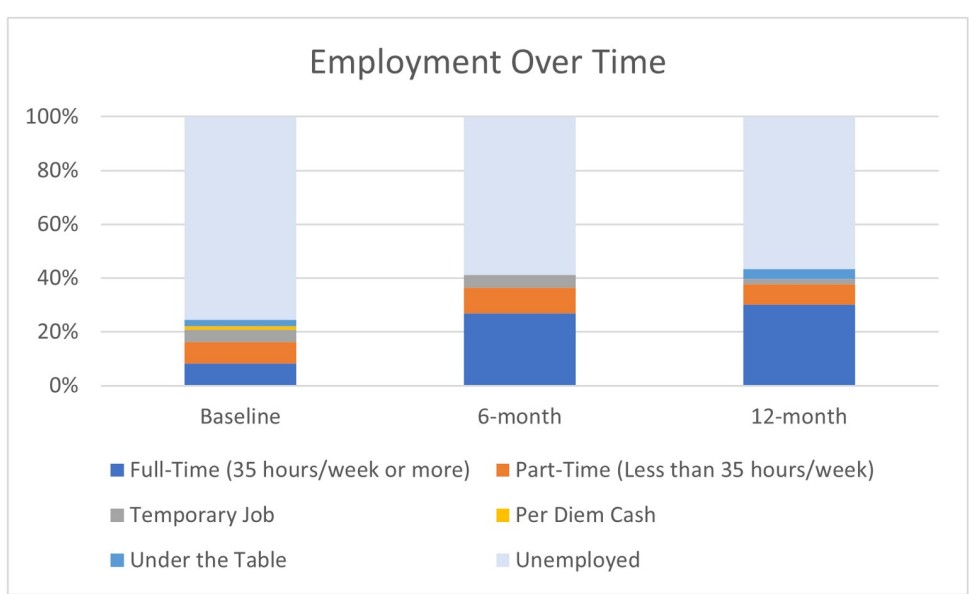

**Fig 2. Employment.** At each subsequent measurement wave, participants were approximately 1.9 times ($p<0.001$) more likely to be in a better employment status.

significant, positive predictor of viral suppression. At each subsequent measurement wave, study participants were approximately 1.6 times ($p<0.05$) more likely to be virally suppressed.

### Retention in care

Retention in care did not significantly improve for participants in the intervention. The model analyzing retention in care by measurement wave (S5 Table) indicated that time was not a significant predictor of retention (OR = 0.820; $p$ = 0.370). Although study participants were approximately 18% less likely to be retained in care with each subsequent wave, this change was not statistically significant.

## Discussion

The KC Life 360 intervention to improve housing and employment for at-risk, low-income PLWH shows promising results related to improving housing, employment, and ultimately improving HIV health outcomes. Most participants of this study were able to achieve a more stable housing situation than at baseline, with nearly 75% of people reporting stable housing after one year. These results are supported by other studies that suggest client navigation may be a potential intervention to improve housing [10, 11].

The literature on navigation to improve employment is less robust. In this intervention, employment was more difficult to improve than housing. Nearly half of participants reported some kind of employment after one year, with roughly 1/3 reporting full-time employment. The results of this study support the use of client navigation to improve employment. For some, this could be related to structural factors such as the availability of jobs in the Kansas City area. However, future studies need to identify which aspects of client navigation are most important to improve employment, versus housing status.

Viral suppression significantly improved over the intervention period. It seems that KC Life 360 may have positive impacts on HIV health outcomes in one year. These promising results provide evidence that client navigation may be able to improve viral suppression in a relatively short time. The authors of this study suggest that these important results should be viewed in context of the limitations. Retention in care did not significantly change over the intervention period. There were a large number of individuals who were not virally suppressed at baseline. Some of these same individuals were not retained in care from baseline to follow-up. While not statistically different, it seems that those individuals who were virally suppressed and/or retained in care at baseline were retained more in the intervention over time. This is expected, as those who have relationships with healthcare providers are more likely to maintain those relationships [23]. While we examined the differences in demographic and psychosocial variables listed in Table 1, we found no differences between groups that would help clarify why some individuals were retained in this intervention versus others. Future research needs to be conducted on the mechanisms that may increase retention and decease attrition in this population.

This intervention used a two-pronged approach: client navigation and system-level service coordination. The results of this study did not examine the degree to which system-level service coordination may have impacted client-level outcomes. It is possible that client navigation was aided by the system-level intervention factors. Lightner, et al. and Prochnow, et al have shown that service coordination is related to the social networks of providers [24, 25]. To date, no studies have been conducted on how changing system-level factors are related to client-level outcomes such as improved housing and employment. Our multisectoral team met weekly to discuss client cases related to housing and employment and connect clients to resources including job announcements and available apartments. Collecting data on the types

of activities and length of time to get a client housed or employed is needed. The field should continue to examine how incorporating employment and housing services with medical care may improve the lives of PLWH.

This study is strengthened by examining housing and employment for a group of low-income, traditionally underserved, PWLH experiencing housing instability in the Ryan White system. The use of objective medical chart data for viral load and retention in care, and the one-year intervention period adds additional strength to this study. However, this study lacks generalizability to other populations who may have access to additional resources. Additionally, due to the multiple methods of data collection (interviews and medical charts), some follow-up data are missing. For example, a participant could meet with their medical provider but not be interviewed by study staff. Our sample was from a mid-size US metropolitan area and may not be relevant for people with HIV from rural areas. A key limitation of this study is the lack of a comparison group thus limiting the generalizability of the results. Future research and funding should focus on incorporating a comparison group to determine if the intervention potentially caused additional improvements in health and wellbeing beyond what would occur without the intervention. Additionally, due to the high rate of attrition, results should be viewed in the context of those who are able to maintain contact with providers over time.

Housing and employment navigation could be an effective tool for providers who serve PLWH most in need of services. Future research needs to be conducted on the potential reproducibility of these results in other areas and scalability of interventions like KC Life 360 on housing and employment for highly marginalized populations. Additionally, long-term projects should be conducted to understand the potential multi-year impact of housing and employment navigation on the lives of PLWH.

## Supporting information

**S1 File. Data release.**
(XLSX)

**S1 Table. Results from ordinal logistic GEE for housing.**
(DOCX)

**S2 Table. Results from ordinal logistic GEE for employment.**
(DOCX)

**S3 Table. Results from binary logistic GEE for employment.**
(DOCX)

**S4 Table. Results from binary logistic GEE for viral suppression.**
(DOCX)

**S5 Table. Results from binary logistic GEE for retention in care.**
(DOCX)

## Acknowledgments

First, we would like to thank the dedication and hard work of our partners in this intervention. Specifically, Christy Rodman, Traquel Harrison, Stephanie Boyer, Jonathan Roberts, Danna Stone, Whitney Hutchinson, Desiree Blake, Erica Helin, and Hanni Woelk from ReStart Inc for providing housing support and Jessica Gant, Simon Muturi, Kathy Vereecke-Ficcadenti, and Dionne Blake from Catholic Charities of Kansas City and St. Joseph for providing employment support.

The Health Resources and Services Administration's (HRSA) Ryan White HIV/AIDS Program (RWHAP) provides a comprehensive system of HIV primary medical care, essential support services, and medications for low-income people with HIV who are uninsured and underserved. The Program funds grants to states, cities/counties, and local community-based organizations to provide care and treatment services to people living with HIV to improve health outcomes and reduce HIV transmission among hard-to-reach populations.

## Author Contributions

**Conceptualization:** Joseph S. Lightner, Jamie Shank, Serena Rajabiun.

**Data curation:** Joseph S. Lightner, Travis Barnhart, Jamie Shank, Debbie Adams.

**Formal analysis:** Steven Chesnut.

**Funding acquisition:** Jamie Shank.

**Investigation:** Joseph S. Lightner, Jamie Shank, Ella Valleroy, Serena Rajabiun.

**Methodology:** Joseph S. Lightner, Ella Valleroy, Serena Rajabiun.

**Project administration:** Debbie Adams, Serena Rajabiun.

**Software:** Joseph S. Lightner.

**Supervision:** Joseph S. Lightner, Travis Barnhart, Jamie Shank, Serena Rajabiun.

**Validation:** Joseph S. Lightner, Serena Rajabiun.

**Visualization:** Joseph S. Lightner.

**Writing – original draft:** Joseph S. Lightner, Steven Chesnut, Serena Rajabiun.

**Writing – review & editing:** Joseph S. Lightner, Travis Barnhart, Jamie Shank, Ella Valleroy, Steven Chesnut, Serena Rajabiun.

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
