## [Decision Letter · Decision Letter 0]

22 Jul 2022

PONE-D-22-12170Outcomes of the KC Life 360 Intervention: Improving employment and housing for persons living with HIVPLOS ONE

Dear Dr. Lightner,

Thank you for submitting your manuscript to PLOS ONE. After careful consideration, we feel that it has merit but does not fully meet PLOS ONE’s publication criteria as it currently stands. Therefore, we invite you to submit a revised version of the manuscript that addresses the points raised during the review process.

While the reviewers agree that this is an important topic, there are some issues with the paper in its current form that need to be addressed.  Please see below for the reviewers' comments.

We look forward to receiving your revised manuscript.

Kind regards,

Bettye A. Apenteng

Academic Editor

PLOS ONE

Journal Requirements:

Reviewers' comments:

Reviewer's Responses to Questions

**Comments to the Author**

1. Is the manuscript technically sound, and do the data support the conclusions?

Reviewer #1: Partly

Reviewer #2: Yes

2. Has the statistical analysis been performed appropriately and rigorously? 

Reviewer #1: I Don't Know

Reviewer #2: Yes

3. Have the authors made all data underlying the findings in their manuscript fully available?

Reviewer #1: Yes

Reviewer #2: Yes

4. Is the manuscript presented in an intelligible fashion and written in standard English?

Reviewer #1: Yes

Reviewer #2: Yes

5. Review Comments to the Author

Reviewer #1: Generally, this is an important analysis because it describes the outcomes for a particularly vulnerable population in an underrepresented region of the US. However, the results are stated overly strongly and should be tempered by some discussion of how the lack of a comparison group limits generalizability.

Abstract:

Are the % changes over baseline for all outcomes? Please include or describe in a standardized way. Please add the p values in the abstract.

Introduction:

Last paragraph in the introduction should outline more specifically what the relationships are under study in this analysis.

Methods:

The sections included are not really standard and the reasons for the way they are ordered is a little unclear. Is recruitment the same as enrollment in the study? It should probably be grouped with data collection and measures, which are separate from the intervention description section. Suggest reordering.

Participants: The number listings in the participants paragraph are not needed. The definitions of homeless (literally, imminent, etc.) should be specified.

Analysis: It looks like time is the proxy for the intervention because there’s no comparison group? This should be stated more clearly.

Results:

The table names could be more specific, for example Table 1 might be better described as “Participant characteristics.” Table 2 should be “Frequency and percent of patients reporting outcomes” or something like that…

The changes in clinical indicators might be better presented as figures in change over time?

Are there statistical tests available across categories of outcomes? It looks like the supplemental tables have most of the statistical test, which should be in the paper instead. As an alternative, stat sig could be indicated in each category of housing in figure 1, for example.

Representing the results with a more traditional table/s of regression results would strengthen the paper.

Discussion

The discussion argues that people achieved better outcomes than “before” the intervention, but there is no comparison data available for prior periods, just a baseline measure so the comparison is baseline to follow up, not really “before.” Similarly, is there any information on how many people achieve housing in a year, regardless of navigation? There must be some secular trend and the lack of a comparison group makes it difficult to assess how much of the change here is from that. Same goes for viral suppression so many of the findings are tentative without a comparison group and should be stated less strongly. The lack of a comparison group (contemporaneous or prior time period) is a key limitation and should be specified in the discussion.

Reviewer #2: This is an important study, and is well presented. Given that the study primarily addresses employment and housing, it would be useful for the authors to state whether the the navigation components include assistance with 'maintaining' housing and employment and the frequency of meetings with the navigators after housing and employment was obtained. Although included in the Discussion a 'Conclusion' section would be meaningful.

Recommendations below are primarily to improve clarity, and readability.

Suggested Edits:

40: Add 'being' to reporting ...

50: Clarify whether gift cards were given 'during' or at completion of interview?

65: All 'Americans' or to be more accurate, US residents or US population?

68-69: Were meetings discontinued after clients were stably housed?

97: Consider deleting 'significantly'.

135-136: Not sure if this is a standard definition; if not, please clarify "not having a job that used all their skills".

138-139: Recommend using a multilevel numbering system for the HIV health risk factors; e.g. lower case alphabets (a,b c) or roman numerals (i, ii, iii) if allowed.

158: Add documents to 'identification'.

183-185: Clarify - were case notes used to record current status/progress?

183: Clarify whether documentation was on a shared system.

186-187: What does local reporting mean i.e. who had access?

209, 211: Delete 'has'

223: Consider changing to : 'was assessed at baseline to capture the last primary care visit ...'

250-252: Although listed under references, consider naming the scales used to assess depression and food insecurity.

253: Consider listing all 12 unmet needs so that there is alignment with what you present in results.

254: Change of to 'for' after need.

261-269: Change '<' symbols to commas?

309: Change 'were' to 'was'.

320: Previous sentence states viral suppression increased but sentence states retention in care 'also' decreased. This does not align.

323: Consider specifying 'change', i.e. replace with 'improved', 'increased', 'decreased', etc.

327: Change individual to individuals, i.e. singular to plural.

365: Consider replacing 'at' with 'related to'.

376: Add 'use of' client navigation to ...

379: Consider replacing 'compared to' with 'versus'.

383: Clarify direction of change.

386: Change 'was to 'were'.

386-387: Confusing - please clarify.

392: Clarify - 'no differences' in variables or the impact of those variables?

397: Clarify - consider 'this study does not present of did not examine'.

400: Clarify - 'authors' or 'study'.

Table 1: Clarify Food Security scores. In line 253 it is stated that higher scores indicate more food insecurity in which case it would be appropriate to title this as 'Food Insecurity'.

Table 2: a) Explain discrepancy in numbers between various categories; b) Either include totals for each category or remove them.

6. PLOS authors have the option to publish the peer review history of their article (what does this mean?). If published, this will include your full peer review and any attached files.

Reviewer #1: No

Reviewer #2: No

---

## [Author Response · Author response to Decision Letter 0]

1 Aug 2022

The responses to all comments are addressed in the attached file.

---

## [Decision Letter · Decision Letter 1]

1 Sep 2022

PONE-D-22-12170R1Outcomes of the KC Life 360 Intervention: Improving employment and housing for persons living with HIVPLOS ONE

Dear Dr. Lightner,

Thank you for submitting your manuscript to PLOS ONE. After careful consideration, we feel that it has merit but does not fully meet PLOS ONE’s publication criteria as it currently stands. Therefore, we invite you to submit a revised version of the manuscript that addresses the points raised during the review process.

Specifically, the comments raised by Reviewer # 2, which are fairly minor in my assessment, need to be addressed. I believe these changes can be made fairly quickly.

We look forward to receiving your revised manuscript.

Kind regards,

Bettye A. Apenteng

Academic Editor

PLOS ONE

Journal Requirements:

Reviewers' comments:

Reviewer's Responses to Questions

**Comments to the Author**

1. If the authors have adequately addressed your comments raised in a previous round of review and you feel that this manuscript is now acceptable for publication, you may indicate that here to bypass the “Comments to the Author” section, enter your conflict of interest statement in the “Confidential to Editor” section, and submit your "Accept" recommendation.

Reviewer #1: All comments have been addressed

Reviewer #2: (No Response)

2. Is the manuscript technically sound, and do the data support the conclusions?

Reviewer #1: Yes

Reviewer #2: Yes

3. Has the statistical analysis been performed appropriately and rigorously? 

Reviewer #1: Yes

Reviewer #2: Yes

4. Have the authors made all data underlying the findings in their manuscript fully available?

Reviewer #1: Yes

Reviewer #2: Yes

5. Is the manuscript presented in an intelligible fashion and written in standard English?

Reviewer #1: Yes

Reviewer #2: Yes

6. Review Comments to the Author

Reviewer #1: The authors addressed all of my comments. The addition of the language around the lack of a comparison group is particularly helpful and strengthens the paper.

Reviewer #2: Authors have addressed previous comments, however the following guidance should be considered to improve accuracy,clarity and readability.

General: a) Please check for punctuation, spelling and grammar; and b) Be specific when referring all PLWH vs study participants.

103: Delete 'to'

111: Replace 'be related to' with 'help promote'

121, 122: Delete 'who'

136: Replace 'received' with 'obtained'

138: Replace 'of' with 'related to'

142: Clarify 'emergency assistance'

145: Combine 2nd sentence to preceding sentence stating 'and continued to to meet with them as needed after they achieved hosing stability'.

150: Replace 'offices' with 'service sites'

175-181: Review punctuation and use of capital letters to help with clarity

200: Change capital 'T' to lower case in 'temporary'

219: Change to 'through lab data obtained from'

224: Change 'were' to 'was'

228: Change 'assessing' to 'assessed'

229: add 'recorded' in

232: Add as 'being' out ...

269-276: Add '=' after each number e.g. (1)=literally ...

294: Add 'from study participants'

300: Replace 'were' to 'was'

307: Replace 'was' with 'were'

305, 307: Re-state 'average individual' to more accurately reflect results/analysis

317: Change 'were' to 'was'

338: Change PLWH to 'study participants'

339: Change to - 'to have improved housing status'

382: Clarify and specify - did a large proportion of study participants need additional support to get employment or to stay employed?

387: 'Housing' or 'housing status'?

393: Context of?

402: Delete extra period

403: attrition as well as retention?

407: Change to 'the study did not examine' or 'the results do not explain'

426: Change data 'is' to 'are'

427: Interviewed by whom?

429,430: Change to 'thus limiting' the ...

431: Replace 'understand' with 'determine'

438: Should this be 'and scalability of' ...

7. PLOS authors have the option to publish the peer review history of their article (what does this mean?). If published, this will include your full peer review and any attached files.

Reviewer #1: No

Reviewer #2: No

---

## [Editor Report · Decision Letter 2]

7 Sep 2022

Outcomes of the KC Life 360 Intervention: Improving employment and housing for persons living with HIV

PONE-D-22-12170R2

Dear Dr. Lightner,

We’re pleased to inform you that your manuscript has been judged scientifically suitable for publication and will be formally accepted for publication once it meets all outstanding technical requirements.

Kind regards,

Bettye A. Apenteng

Academic Editor

PLOS ONE
---

## [Editor Report · Acceptance letter]

8 Sep 2022

PONE-D-22-12170R2 

Outcomes of the KC Life 360 Intervention: Improving employment and housing for persons living with HIV 

Dear Dr. Lightner:

I'm pleased to inform you that your manuscript has been deemed suitable for publication in PLOS ONE. Congratulations! Your manuscript is now with our production department. 

Kind regards, 

on behalf of

Dr. Bettye A. Apenteng 

Academic Editor

PLOS ONE